# Integrated Metabolome and Lipidome Strategy to Reveal the Action Pattern of Paclobutrazol, a Plant Growth Retardant, in Varying the Chemical Constituents of Platycodon Root

**DOI:** 10.3390/molecules27206902

**Published:** 2022-10-14

**Authors:** Lan Lan, Weizhen Huang, Heng Zhou, Jiajia Yuan, Shui Miao, Xiuhong Mao, Qing Hu, Shen Ji

**Affiliations:** 1NMPA Key Laboratory for Quality Control of Traditional Chinese Medicine, Shanghai Institute for Food and Drug Control, Shanghai 201203, China; 2School of Pharmacy, Yantai University, Yantai 264005, China

**Keywords:** platycodon root, paclobutrazol, metabolomic and lipidomic differences, quantitative analysis, plant growth regulators

## Abstract

Platycodon root, a medicinal food homology species which has been used in Asian countries for hundreds of years, is now widely cultivated in China. Treatment with paclobutrazol, a typical plant growth retardant, has raised uncertainties regarding the quality of Platycodon root, which have been rarely investigated. In the present study, metabolomic and lipidomic differences were revealed by ultra-high performance liquid chromatography coupled to ion mobility-quadrupole time of flight mass spectrometry (UPLC-IM-QTOF-MS). A significant decrease of platycodigenin-type saponins was observed in the paclobutrazol-treated sample. Carrying out a comprehensive quantitative analysis, the contents of total saponins and saccharides were determined to illustrate the mode of action of paclobutrazol on Platycodon root. This study demonstrated an exemplary research model in explaining how the exogenous matter influences the chemical properties of medicinal plants, and therefore might provide insights into the reasonable application of plant growth regulators.

## 1. Introduction

Platycodon root, the dried root of *Platycodon grandiflorus* (Jacq.) A.DC. (Campanulaceae), is an ingredient in many traditional medicines that are clinically used in the relief of respiratory diseases such as bronchitis, asthma, tonsillitis, and pulmonary tuberculosis [1,2,3]. Furthermore, it has been utilized at the forefront of the battle against COVID-19 in China, as it possesses remarkable effects in alleviating symptoms, improving cure rate, reducing death rate and promoting recovery [4]. According to a statistical analysis, among the 193 varieties of botanical medicines used in 141 representative formulas and Chinese Patent Medicines, the usage frequency of Platycodon root is ranked as 10th [5]. The importance of Platycodon root as a medicinal herb is increasing with the awareness of public health.

Platycodon root is also a medicine food homology species and is commonly consumed as a salad and preserved vegetable in the northeast of China, Japan and Korea [6,7]. According to the general administration of customs of China, the trade of Platycodon root is climbing and this product was exported to many countries in 2021, such as South Korea, Japan, Malaysia, Thailand, Russia, Canada and the Netherlands. It is of huge economic benefit for herbal farmers to cultivate Platycodon root as an agricultural product. Consequently, the utilization of exogenous plant growth regulators (PGRs) has occurred in agricultural activities to increase its output. Exogenous PGRs are several categories of compounds that share similar physiological and biochemical reactions with phytohormones in affecting plant life progress from seed germination to fruit abscission. Of these organic agents, plant growth retardants are important species which can inhibit cell division, elongation and growth rate by regulating plant stem tip sub-apical meristem cells or primordial meristem cells [8].

Paclobutrazol is one of the most used plant growth retardants in planting management; it belongs to the endogenous gibberellin synthesis inhibitor group and is highly effective and low-cost [9]. Its application scope in China under pesticide registration covers rice, wheat, peanut and varieties of fruit trees. However, the utilization of paclobutrazol extends to the cultivation of radix and rhizome herbal medicines such as *Codonopsis Radix* [10], *Panax ginseng* [11] and *Ophiopogonis Radix* [12,13]. This has spread to Platycodon root recently, but the influence of paclobutrazol on the chemical constituents of Platycodon has not been investigated. PGRs alter cellular responses of cells and further impact the synthesis and accumulation of primary and secondary metabolites. Levels of key bioactive compounds fluctuate with different patterns in various herbal species. The research on different varieties needs to be expanded and the scope of concerned compounds should be widened.

Previous studies have shown that triterpenoid saponins [14,15] and lipids [16,17] are the target compounds that have attracted the most interest. Both nutritional and pharmaceutical values are taken into consideration from metabolome and lipidome aspects. This study provides a new insight in figuring out the mechanisms of paclobutrazol on a typical traditional Chinese medicine (TCM) product and in promoting the reasonable use of PGRs.

## 2. Results and Discussion

### 2.1. Effects of Paclobutrazol on the Growth of Platycodon Root

The alteration of fruit shape and production yield have been investigated in response to paclobutrazol application in food production [18,19], but its influence on herbal medicinal plant components is less studied. In this study, ten Platycodon roots were handpicked for each group. There was little difference in the appearance of control and paclobutrazol-treated samples, with the whole roots being yellowish white, cylindrical or slightly fusiform, gradually tapering downwards, sometimes branched, slightly twisted and with several scars on the taproot. Root diameter and weight were compared to further demonstrate the influence of paclobutrazol (Figure 1). An obvious weight gain was observed for the paclobutrazol-treated group. For 2-year-old root samples, paclobutrazol treatment increased the diameter by 16.32% (from 2.39 ± 0.53 cm to 2.78 ± 0.45 cm) and the weight by 39.42% (from 41.38 ± 9.72 g to 57.69 ± 15.36 g). After application of paclobutrazol, 3-year-old root samples increased by 13.98% (from 3.22 ± 0.61 cm to 3.67 ± 0.30 cm) in diameter and 40.90% (from 62.81 ± 20.58 g to 88.50 ± 25.75 g) in weight.

### 2.2. Effect of Paclobutrazol on Metabolic Profile of Platycodon Root

Orthogonal partial least squares discriminant analysis (OPLS-DA) was conducted with 70% methanol extracts of 2-year-old and 3-year-old Platycodon roots. As shown in Figure 2, the OPLS-DA score plot showed grouping trends between control and paclobutrazol-treated samples. In the OPLS-DA models, R^2^Y and Q^2^ were 0.997 and 0.983 in 2-year-old samples and 0.997 and 0.982 in 3-year-old samples, respectively. Two hundred permutations were performed to validate the models, which exhibited good predictability without data overfitting.

Variable importance in the projection (VIP) was calculated using a nonparametric test and differential metabolites were exposed by the threshold of VIP > 1 and *p* < 0.05. Data correction, peak picking, and peak annotation were performed through UNIFI 1.9.4.0 software. The identification of compounds was first matched using an *in-house* database with the compounds collected from Full-Text Database (CNKI), SciFinder and Reaxys [16]. Further verification and reasonable structural elucidation based on reference standards, chromatographic elution behaviors and online searching were completed. Collectively, 26 differential metabolites in 2-year-old Platycodon roots were found, with the relative contents of 20 metabolites, including 10 triterpenoid saponins, 5 saccharides and 5 fatty acids, showing a decreasing trend for the paclobutrazol-treated group compared with the control (Figure 2c). Similarly, 22 differential metabolites in 3-year-old Platycodon roots were discovered with levels of five triterpenoid saponins and one fatty acid decreasing, whereas the contents of seven saccharides and nine triterpenoid saponins increased (Figure 2d). These 38 potential metabolomic markers are summarized in Table 1. Ten metabolomic markers were common in both 2-year-old and 3-year-old experimental samples, involving five oligosaccharide, four saponins and one organic acid. It was reasonably speculated that the contents of certain triterpenoid saponins, saccharides and fatty acids might fluctuate with the application of paclobutrazol, most probably in a complicated pattern.

Saponins are widely distributed in herbal medicines with vast structural and functional diversity [20,21,22]. As the most-studied bioactive secondary metabolites, saponin derivatives in Platycodon root can be roughly divided into five types according to the aglycone group: platycodigenin, polygalacic acid, platycogenic acid, platycogenic acid lactone type and others [23,24]. To further reveal the action mode of paclobutrazol, the abundance of differential metabolites belonging to particular saponin types was investigated. Trends were found where platycodigenin-type saponins decreased after the treatment, while the polygalacic-acid-type saponins were significantly higher (*p* < 0.001) than the control in 2-year-old roots (Figure 3).

**Table 1 molecules-27-06902-t001:** Potential metabolomic markers for Platycodon root with different cultivation years and paclobutrazol treatment.

NO.	Compound ID	Adducts	Formula	Mass Error/ppm	MS/MS Information	Identify	Treatments
1	10.53_1279.5597*m*/*z*	[M−H]^−^	C59H92O30	−0.25	1127.4875, 695.3644, 679.3696, 519.3330, 469.1564	3″-*O*-acetylplatyconic acid A	2Y
2	11.09_1121.5372*m*/*z*	[M+HCOO]^−^	C52H84O23	−1.23	1075.5308, 753.4072, 681.3852, 665.3905, 541.1770, 469.1563	platycoside J	2Y
3	11.37_1473.6372*m*/*z*	[M+HCOO]^−^	C65H104O34	−1.31	1385.6228, 1367.6129, 843.4386, 825.4280, 781.4382, 663.3752	3″-*O*-acetyl platycodin D2	2Y
4	11.56_1266.5938*n*	[M−H]^−^	C59H94O29	−0.38	1223.5702, 723.3969, 681.3851, 663.3754, 541.1772, 469.1565	platycodin C	2Y, 3Y
5	11.95_1457.6422*m*/*z*	[M+HCOO]^−^	C65H104O33	−0.26	1369.6268, 827.4436, 809.4331, 765.4435, 647.3804, 541.1777	3″-*O*-acetylpolygalacin D2	2Y, 3Y
6	12.49_1279.5598*m*/*z*	[M−H]^−^	C59H92O30	−0.18	1249.5525, 1069.4838, 695.3647, 485.2907, 471.3113, 423.2904	platycodin L	2Y
7	12.79_1279.5594*m*/*z*	[M−H]^−^	C59H92O30	−0.52	995.4454, 849.3937, 717.3456, 695.3647, 469.2963, 409.3110	2″-*O*-acetylplatyconic acid A	2Y
8	13.74_1473.6371*m*/*z*	[M+HCOO]^−^	C65H104O34	−1.43	1367.6112, 843.4381, 825.4280, 781.4389, 663.3743, 519.3329	2″-*O*-acetyl platycodin D2	2Y
9	14.04_1266.5937*n*	[M−H]^−^	C59H94O29	−0.05	1223.5674, 723.3954, 681.3850, 541.1770, 469.1561	platycodin A	2Y, 3Y
10	14.55_1457.6424*m*/*z*	[M+HCOO]^−^	C65H104O33	−1.30	1369.6276, 1147.5481, 827.4432, 809.4327, 647.3801, 491.1380	2″-*O*-acetylpolygalacin D2	2Y, 3Y
11	15.47_1279.5586*m*/*z*	[M−H]^−^	C59H92O30	−1.16	1249.5469, 1069.4835, 695.3639, 665.3540, 633.3631, 471.3111	platycodin K	2Y
12	19.54_519.3323*m*/*z*	[M−H]^−^	C30H48O7	−0.84	473.3248, 457.3316, 407.2954, 341.2212, 313.2396	platycodigenin	2Y
13	7.11_1532.6860*n*	[M+HCOO]^−^	C69H112O37	−1.47	1531.6792, 1239.5618, 989.4940, 665.3904, 541.1773, 469.1559	platycoside D	2Y
14	7.71_1254.5934*n*	[M+HCOO]^−^	C58H94O29	−1.08	1253.5785, 843.4378, 825.4278, 681.3845, 519.3325, 471.3116	deapioplatycodin D3	2Y
15	2.15_828.2724*n*	[M+HCOO]^−^	C30H52O26	−2.72	827.2669, 647.2034, 485.1507, 341.1086, 323.0979, 179.0559	1-fructofuranosylnystose	2Y, 3Y
16	3.25_1314.4321*n*	[M+HCOO]^−^	C48H82O41	−0.81	1313.4254, 1133.3627, 971.3085, 809.2560, 647.2036, 161.0457	1,1,1,1,1,1-kestooctaose	2Y, 3Y
17	3.33_1197.3771*m*/*z*	[M+HCOO]^−^	C42H72O36	−1.20	1151.3725, 971.3081, 827.2674, 647.2037, 342.1127, 179.0562	1,1,1,1,1-kestoheptaose	2Y, 3Y
18	3.33_1476.4839*n*	[M+HCOO]^−^	C54H92O46	−1.38	1475.4770, 1151.3725, 989.3190, 809.2569, 665.2148, 557.1721	1,1,1,1,1,1,1-kestononaose	2Y, 3Y
19	3.33_990.3318*n*	[M−H]^−^	C36H62O31	−0.95	809.2565, 647.2037, 485.1510, 395.1200, 323.0982, 179.0562	1,1,1,1-kestohexaose	2Y, 3Y
20	6.71_441.1765*m*/*z*	[M+HCOO]^−^	C20H28O8	−0.40	395.1709, 305.1243, 215.1082, 185.0976, 159.0817, 143.0708	lobetyolin	2Y
21	22.76_433.2356*m*/*z*	[M−H]^−^	C21H39O7P	−1.00	433.2361, 279.2329, 152.9961	LPA 18:2	2Y
22	22.61_295.2280*m*/*z*	[M−H]^−^	C18H32O3	0.50	277.21745, 233.2285, 195.1394, 183.1024, 125.0952	9-hydroxy-10,12-octadecadienoic acid or isomer	2Y, 3Y
23	23.01_293.2120*m*/*z*	[M−H]^−^	C18H30O3	−0.64	275.2026, 249.2222, 195.1391, 167.1086, 139.1137, 113.0974	9-oxo-10(E),12(E)-octadecadienoic acid or isomer	2Y
24	25.43_277.2173*m*/*z*	[M−H]^−^	C18H30O2	−0.02	259.2069, 233.2278, 221.1551, 209.1552	linolenic acid	2Y
25	26.21_279.2331*m*/*z*	[M−H]^−^	C18H32O2	0.47	279.2332, 261.2231, 233.1901	linoleic acid	2Y
26	26.62_255.2328*m*/*z*	[M−H]^−^	C16H32O2	−2.18	255.2336, 238.2253	palmitic acid	2Y
27	11.04_1223.5691*m*/*z*	[M−H]^−^	C57H92O28	−0.57	1091.5254, 681.3850, 663.3751, 635.3793, 519.3329, 469.1564	platycodin D	3Y
28	10.92_1386.6299*n*	[M−H]^−^	C63H102O33	−0.08	843.4389, 825.4283, 781.4385, 663.3751, 541.1780, 469.1566	platycodin D2	3Y
29	12.35_1250.5932*n*	[M+HCOO]^−^	C59H94O28	−0.70	1249.5852, 1207.57340 665.3907, 469.1565, 409.1353	3″-*O*-acetylpolygalacin D	3Y
30	15.20_1250.5932*n*	[M−H]^−^	C59H94O28	−1.25	1207.5743, 665.3903, 469.1563, 665.3903, 469.1563, 409.1351	2″-*O*-acetylpolygalacin D	3Y
31	12.04_1237.5497*m*/*z*	[M−H]^−^	C57H90O29	−0.21	1207.5386, 1027.4747, 695.3643, 665.3541, 471.3116, 423.2906	platycodin J	3Y
32	13.30_1325.6009*m*/*z*	[M+HCOO]^−^	C60H96O29	−1.27	1279.5970, 827.4432, 809.4326, 765.4430, 485.3282	deapi-2″-*O*-acetyl-polygalacin D2	3Y
33	11.23_1325.6011*m*/*z*	[M+HCOO]^−^	C60H96O29	−0.56	1325.6012, 1237.5801, 827.4433, 809.4328, 765.4435,	deapi-3″-*O*-acetyl-polygalacin D3	3Y
34	11.43_1370.6347*n*	[M−H]^−^	C63H102O32	0.09	1045.5170, 941.4794, 827.4432, 737.4197, 665.3890, 469.1566	polygalacin D2	3Y
35	8.26_1427.6331*m*/*z*	[M+HCOO]^−^	C65H104O34	−0.41	1385.6227, 1367.6132, 843.4382, 825.4271, 519.3330, 469.1565	3″-*O*-acetyl platycodin D3	3Y
36	9.64_1428.6403*n*	[M+HCOO]^−^	C65H104O34	−0.97	1385.6215, 1367.6215, 843.4380, 825.4275, 519.3324, 469.1565	2″-*O*-acetyl platycodin D3	3Y
37	1.14_666.2211*n*	[M+HCOO]^−^	C24H42O21	−0.51	711. 2197, 665.2143, 485.1514, 341.1095, 323.0981,179.0560	nystose	3Y
38	0.46_341.1082*m*/*z*	[M−H]^−^	C12H22O11	0.09	272.0881, 221.0659, 179.0565, 161.0456	sucrose	3Y

**Figure 3 molecules-27-06902-f003:**
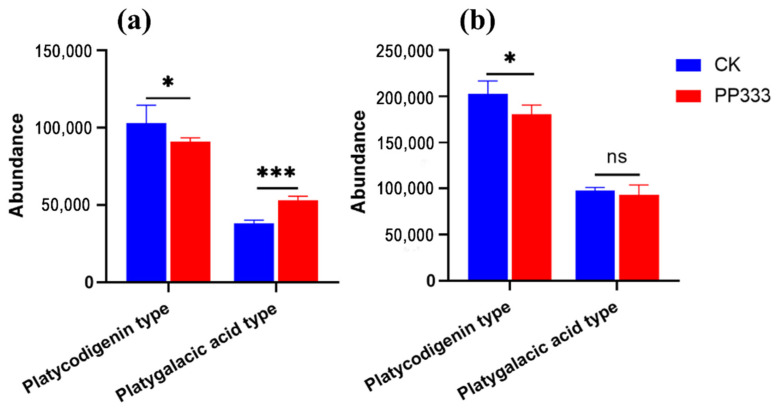
Level of platycodigenin-type and platygalacic-acid-type saponins in 2-year-old (**a**) and 3-year-old (**b**) Platycodon roots. Data are presented as mean ± SD (*n* = 6). Significant differences are shown as * *p* < 0.05, and *** *p* < 0.001.

### 2.3. Effect of Paclobutrazol on Metabolic Profile of Platycodon Root

In the lipidomic analysis, OPLS-DA prediction models were well established in both positive and negative ionization mode (Figure 4). The control and paclobutrazol-treated groups showed obvious trends of intra-group aggregation and inter-group segregation, suggesting that paclobutrazol significantly changed the lipid profile in Platycodon roots. Permutation tests, VIP plots and heatmaps visualizing the relative content differences are shown in Appendix A. Based on a screening threshold of VIP > 1 and *p* < 0.05, a total of 116 differential lipids were selected from the control and paclobutrazol-treated groups of 2- and 3-year-old Platycodon roots (Appendix A). It was remarkable that the in-house components library for Platycodon root developed previously assisted in the quick identification of the differential chemicals [16,17].

Lipids are one of the important nutrients in Platycodon roots. Technical iterations have promoted the lipidic development of quantitation and high throughput methods. Total lipids of six classes were calculated using isotope-labeled internal standards (ISs) relying on the MS-based response of the corresponding peaks [25,26]. Six lipid standards including 15:0−18:1 (*d*7) PC (for quantification of PC), 15:0−18:1 (*d*7) PE (for quantification of PE, Cer and PMeOH), 16:0−18:1 (*d*31) PA (for quantification of PA), 15:0−18:1 (*d*7) PS (for quantification of PS), 15:0−18:1 (*d*7) DG (for quantification of DG), and 15:0−18:1 (*d*7) −15:0 TG (for quantification of TG) were used to relatively quantify eight subclasses of lipids in both Platycodon root varieties. It must be explained that 15:0−18:1 (*d*7) PE served as the reference since internal standards of some lipid categories were not available.

As shown in Figure 5, PE, PC, TG and DG increased in both 2-year-old and 3-year-old samples after the paclobutrazol treatment, while Cer decreased significantly (*p* < 0.001). The treatment did not affect PS in 2-year Platycodon roots, but it increased in the 3-year sample. Inconsistent changes were also observed in the lipidomics profile. Levels of PA and PMeOH increased in the 2-year-old sample, but the opposite occurred in the 3-year-old sample. As most previous studies have focused on oil-rich products [27,28], lipidomic analysis for herbal plants would provide new insights in understanding their medicinal and edible value.

### 2.4. Multi-Quantitative Analysis of Key Components in Platycodon Root

#### 2.4.1. Effects of Paclobutrazol on the Content of Total Saponins

Regarding the trends displayed through the metabolome analysis, levels of the platycodigenin-group saponins decreased. A previous study showed that the platycodigenin-type (55.04–68.34%) accounts for the largest proportion of the total saponin content in various parts of *Platycodon grandiflorus* [24]. Thus, key components were selected for further detection by means of the conventional quantitative methods described below.

In this study, the contents of total saponins in the 2-year-old paclobutrazol-treated group by UHPLC-ELSD were significantly lower (*p* < 0.001) compared with the control group, which had values of 0.55% ± 0.02 and 0.72% ± 0.00, respectively (Figure 6a). The content of total saponins for the control group of the 3-year-old sample was 0.83% ± 0.04, whereas the total saponins content declined to 0.64% ± 0.06 after the paclobutrazol treatment, with a significant difference of *p* < 0.05. The results analyzed by UV method were consistent with the above analysis (Figure 6b). This suggested that paclobutrazol might have a negative physiological impact on the biosynthesis and accumulation of saponins in Platycodon roots. A sharp reduction in several principle steroid saponins in paclobutrazol-treated *Ophiopogon japonicus* has been reported [12,13].

#### 2.4.2. Effects of Paclobutrazol on the Content of Polysaccharides and Oligosaccharides

Several studies have shown that saccharides extracted from Platycodon root possess antiviral [29], immunomodulatory, antioxidant, anti-inflammatory and anti-apoptosis effects [30,31,32]. All the effects that might occur after PGR treatment should be taken into consideration, covering a wide range of options.

In this study, saccharides in the Platycodon root were analyzed in two dimensions, i.e., water-soluble polysaccharides and oligosaccharides. However, the contents of these components in 2-year-old and 3-year-old root samples, whether treated with paclobutrazol or not, showed no significant difference. Results are given in Table 2. Liao et al. found that total polysaccharides in *Codonopsis Radix* were reduced by treatment, probably due to an increase in yield [10]. Relevant support for this finding in our study was insufficient; however, the action modalities for different herbs might be different.

## 3. Materials and Methods

### 3.1. Multi-Quantitative Analysis of Key Components in Platycodon Root

Field trials were conducted in 2021 on 2-year-old and 3-year-old Platycodon root samples in Lixing town, Fuyang City, Anhui province (33°26′ N, 115°28′ E), which is the major production area for commercial supply. A 25% paclobutrazol suspension was purchased form Jiangsu Sword Agrochemicals Co., LTD (Yancheng, China). Then, 1200 g/ha of the PGR formulation (300 g/ha active paclobutrazol) was applied twice, with an interval of two weeks, in April for the paclobutrazol-treated group. Planting management was kept the same to the maximum extent until September–October, when the samples of Platycodon root were handpicked during the traditional harvest time in autumn.

### 3.2. Metabolome Analysis

A mixture of 1.0 g fine powder of Platycodon root and 10 mL 70% (*v*/*v*) aqueous methanol was ultrasonically treated in water bath for 30 min at room temperature and then centrifuged at 14,000 r/min. The supernatant was collected for metabolome analysis. QC sample was obtained by equally mixing each test sample and served for the purpose of system real-time stability monitoring. A continuous acquisition was applied and the QC sample was orderly arranged during instrumental analysis.

The experiment was performed on a ACQUITY UPLC system coupled to Vion IMS-QTOF mass spectrometer (Waters Corporation, Milford, MA, USA) equipped with an electrospray ionization (ESI) source. Chromatographic separation was carried out on a Waters CORTECS UPLC^®^ T3 (Waters, Milford, MA, USA) column (2.1 × 100 mm, 1.6 μm) at 40 °C. A binary mobile phase consisting of 0.1% aqueous formic acid (A) and acetonitrile (B) was programed as follows: 0.0–2.0 min, 0–0% B; 2.0–6.0 min, 0–23% B; 6.0–18.0 min, 23–25% B; 18.0–20.0 min, 25–50% B; 20.0–22.0 min, 50–55% B; 25.0–27.0 min, 70–100% B; 27.0–29.0 min, 100% B; and 29.1–33.0 min, 0% B. The flow rate was 0.5 mL min^−1^ and the injection volume was 2 µL.

The MS was conducted in negative ion high-definition MS^E^ (HDMS^E^) mode, and MS parameters were as follows: capillary voltage: 2.0 kV; sample cone voltage: 40 V; source offset voltage: 80 V; source temperature: 120 °C; desolvation temperature: 550 °C; low collision energy: 6 eV; high collision energy ramp: 20–80 eV; cone gas flow rate: 50 L/h; desolvation gas flow rate: 1000 L/h; and analyzer mode: sensitivity. Survey scan data were acquired from *m*/*z* 50 to 2000. Leucine enkephalin was used as the lock mass for both mass and collision cross section (CCS) calibration.

### 3.3. Lipidome Analysis

The lipid extraction procedure was conducted under a methanol–MTBE–water solvent system [33]. Totals of 0.3 mL methanol and 1 mL MTBE were individually added into the 2mL microtube containing 50 mg fine powder of Platycodon root, followed by a 10 min ultrasonic extraction. Phase separation was carried out by adding 0.25 mL water. The upper organic phase was collected and the residue was re-extracted with the same procedure. We then combined the organic phase and concentrated it to a dry state under nitrogen, and then dissolved this in 200 µL methanol. Then, 50 µL of each test sample was pooled together to form the QC sample. The extract was stored at −20 °C before the lipidome analysis. Internal standards were purchased form Avanti Polar Lipids (Alabaster, AL, USA) with the product number of 330709W-1EA (SPLASH^®^ II LIPIDOMIX^®^ Mass spec) and 860453P.

Lipidic data acquisition was conducted on ACQUITY UPLC^®^ CSH column (2.1 × 100 mm, 1.7 μm) at 55 °C with the flow rate of 0.4 mL/min. A binary mobile phase A was acetonitrile/water (60:40, *v*/*v*) with 5mM ammonium formate and 0.1% formic acid, and B was isopropanol/acetonitrile (90:10, *v*/*v*) with 5mM ammonium formate and 0.1% formic acid. Elution gradients were optimized as follows: 0.0–2.0 min, 40–43% B; 2.0–2.1 min, 43–50% B; 2.1–12.0 min, 50–54% B; 12.1–18.0 min, 54–70% B; 12.1–18.0 min, 70–99% B, 18.1–20.0 min, 40% B. The injection volume was 2 µL. Data acquisitions were performed in both positive and negative modes. The MS conditions were the same as above, except for an alternative of 20–40 eV high collision energy ramp.

### 3.4. Assay on Saponins

#### 3.4.1. Total Saponins by UV

A total of 1 g fine herbal powder was weighed accurately and extracted twice with 50 mL water-saturated *n*-butanol for 45 min each time. The organic layer was collected and evaporated to dryness with the residue dissolved into 10 mL with methanol. Then, 200 µL of the test solution was evaporated to dryness in 100 °C water bath. Next, 0.2 mL 5% vanillin-glacial acetic acid and 1.0 mL perchloric acid were added in turn and well-mixed, then heated in a 60 °C water bath for 20 min. We allowed this to cool immediately for 5 min, and 5.0 mL glacial acetic acid was added. Total saponins in Platycodon root were measured 10 min later using an ultraviolet-visible spectrophotometer (Agilent Cary Series UV-Vis, Agilent Technologies, Santa Clara, CA, USA) at absorbance of 474 nm. Calibration curve was built based on the 0.0, 0.1, 0.2, 0.4, 0.6, 0.8 and 1.0 mg/mL Platycodin D bought form national institutes for food and drug controls (NIFDC, Beijing, China) with the same pretreatment as the Platycodon root sample.

#### 3.4.2. Total Saponins by UHPLC-ELSD

The content of total saponins in Platycodon root was determined according to the method recorded in European Pharmacopoeia 10.0 [34]. Chromatographic separation was conducted on an Agilent Poroshell 120 EC-C18 column (4.6 mm × 150 mm, 2.7 µm). A binary mobile phase consisting of water (A) and acetonitrile (B) was programmed in gradient as follows: 0.0–10.0 min, 15–25% B; 10.0–30.0 min, 25% B. The flow rate was 1.0 mL/min, and the injection volume was 10 µL. The temperatures of the column oven and auto-sampler were set as 30 °C and 15 °C, respectively. The evaporator temperature was set to 45 °C and the gas flow for ELSD was set to 1.6 mL/min.

We dissolved 10 mg platycodin D (NIFDC, 111851–201708, purity: 97.9%) with 70% ethanol (water–anhydrous ethanol, 30:70 *v*/*v*) in a 10 mL cliometric flask to obtain the reference solution. We diluted the reference solution to obtain 6 reference solutions, which contained 0.1 mg, 0.2 mg, 0.4 mg, 0.6 mg, 0.8 mg and 1.0 mg platycodin D each per ml. The concentrations should span the expected value in the test solution. Then, we weighed 2.00 g of the powdered herbal drug (through No. 2 sieve), added 50.0 mL 70% ethanol and sonicated for 45 min. This was allowed to cool and was filtered. We rinsed the filter with 10 mL of 70% ethanol. We combined the filtrate and performed washings, and evaporated to dryness under reduced pressure. Then, the residue was taken up with the solvent mixture, transferred to a volumetric flask and diluted to 10.0 mL with the solvent mixture. This was shaken well and filtered through a membrane filter (nominal pore size 0.45 μm).

Then, we established a calibration curve with the common logarithm of the concentration (μg/mL) of reference solutions (corrected by the assigned percentage content of platycodin D) as the abscissa and the common logarithm of the corresponding peak area as the ordinate. We identified the peaks due to 5 saponins by relative retention, which were within ±5% of the specified value. Then we checked the relative retention as follow: saponin 1 = 0.98, saponin 2 (platycodin D) = 1.00, saponin 3 = 1.03, saponin 4 = 1.06, saponin 5 = 1.08. The percentage content of total saponins was calculated, expressed as platycodin D, by taking the sum of the percentage contents of the 5 saponins.

### 3.5. Assay on Saccharides

#### 3.5.1. Water-Soluble Polysaccharides by UV

Phenol-sulfuric acid method was introduced in the determination of water-soluble polysaccharides. An accurately weighed 1 g Platycodon root was ultrasonically extracted in 20 mL water for 30 min. This was allowed to cool. Then, we weighed the stopper conical flask accurately and made up the lost weight with water. Next, 5 mL of the supernatant was partitioned and mixed with 15 mL anhydrous ethanol and allowed to settle overnight. Sediment was obtained by centrifugation at 4000 r/min for 10 min. We rinsed the residue twice with 5 mL ice water, and then dissolved it in a 50 mL flask with water. Next, 200 µL of the test solution was blended with proper amount of water to make it into 1.0 mL. A solution of 1.0 mL 5% phenol was added followed by a quick blend of 5.0 mL sulfuric acid. This was allowed to mix well, then it was heated in a boiling water bath for 15 min. This was allowed to cool immediately for 10 min. The UV-vis spectroscopic method was conducted at absorbance of 486 nm. Calibration curve was built with 0.02, 0.04, 0.08, 0.12, 0.16, 0.20, 0.24, 0.28, 0.32, 0.40 and 0.50 mg/mL anhydrous glucose purchased from NIFDC using the same procedure as the test sample.

#### 3.5.2. Oligosaccharides by UHPLC-ELSD

An aliquot of 0.25 g powder of each Platycodon root sample was weighed accurately and extracted with 25.0 mL of 60% ethanol under ultrasonic conditions for 15 min. This was allowed to cool. We weighed the stopper conical flask accurately and made up the lost weight with 60% ethanol. The flask was shaken and then the sample was filtered through a membrane filter (nominal pore size 0.45 μm). Nystose (NIFDC, 111891–201704, purity: 92.2%) was accurately weighed and dissolved with 60% methanol to prepare stock solution. The stock solution was further diluted to produce a series of standard solutions for calibration curves.

All analyses were performed on an Agilent Series 1260 system (Agilent Technologies, Santa Clara, CA, USA), equipped with evaporative light-scattering detector (ELSD). Chromatographic separation was conducted on a Waters XBridgeTM HILIC (Waters, Milford, MA, USA) column (4.6 mm × 250 mm, 5 µm). A binary mobile phase consisting of water (A) and acetonitrile (B) was programmed in gradient as follows: 0.0–1.0 min, 88% B; 1.0–10.0 min, 88–78% B; 10.0–20.0 min, 78–65% B, 20.0–20.1 min, 65–88% B; 20.1–35 min, 88% B. The flow rate was 1.0 mL/min and the injection volume was 5 µL. The temperature of the column oven was set as 30 °C. The evaporator temperature was set to 45 °C and the gas flow was set to 1.6 mL/min.

We established a calibration curve with the common logarithm of the concentration (μg/mL) of reference solutions (corrected by the assigned percentage content of nystose) as the abscissa and the common logarithm of the corresponding peak area as the ordinate. Then, we identified the peaks due to 7 oligosaccharides by relative retention, which were within ± 5% of the specified value. We checked the relative retention as follows: GF2 = 0.85, GF3 (nystose) = 1.00, GF4 = 1.13, GF5 = 1.25, GF6 = 1.36, GF7 = 1.45, GF8 = 1.53. The percentage content of oligosaccharides was calculated, expressed as nystose, by taking the sum of the percentage contents of GF2-8.

### 3.6. Data Analysis and Quality Assurance

All HDMS^E^ data were collected and processed using UNIFI 1.9 (Waters Corp., Milford, DE, USA) to pass through the apex peak detection and alignment processing algorithms. Normalization of metabolome and lipidome profile data was performed using Progenesis QI 2.1 software (Waters Corporation, Milford, CT, USA). SIMCA v14.1 software (Umetrics, Umea, Sweden) was applied for OPLS-DA. Heatmaps visualizing the relative concentration trends of potential chemical markers was obtained on the web server statistical analysis MetaboAnalyst 5.0 (https://www.metaboanalyst.ca/, access on 20 Jun 2022). Data of the assays were compared and displayed using the statistical software Prism version 8.0 (GraphPad Software, Inc., La Jolla, CA, USA).

Data pretreatment was performed to generate a data matrix that was collected in a continuous acquisition. Raw untargeted metabolomics data were imported into Progenesis QI software, followed by systemic processing of peak alignment, ion fusion, deconvolution, peak picking and normalization. The obtained data matrix was further extracted and screened according to the “30% rule” in QC samples and “80% rule” in test samples before multivariate statistical analysis with SIMCA.

Principal component analysis (PCA) showed that the obtained metabolite and lipid profiling data of QC samples injected into the LC−MS through the whole analytical process were tightly clustered, indicating good instrumental reproducibility and stability in the measurements over the total duration of the experiment. Unsupervised PCA (Appendix A) and supervised OPLS-DA models were performed to investigate the metabolite and lipid variations of Platycodon root.

## 4. Conclusions

In summary, the effect of paclobutrazol on Platycodon root was studied from multiple aspects. Paclobutrazol has complicated impacts on the metabolomic and lipidomic profiles of Platycodon root. It can relatively reduce the content of platycodigenin-type saponins, and at the same time affect the distribution of lipid subclasses. Subsequent quantitative analysis of the key chemical components in Platycodon root showed that paclobutrazol treatment could reduce the content of total saponins, with barely any significant effect on the content of total polysaccharides and oligosaccharides. Our study elucidates the effect of paclobutrazol on the chemical constituents of Platycodon root based on metabolic and lipidomic strategies, which contributes to the medical safety and rational use of plant growth regulators in Chinese herbal medicines. Future research should focus on the molecular mechanisms of the effect of paclobutrazol on the biosynthesis of active ingredients in traditional Chinese medicine.

## Figures and Tables

**Figure 1 molecules-27-06902-f001:**
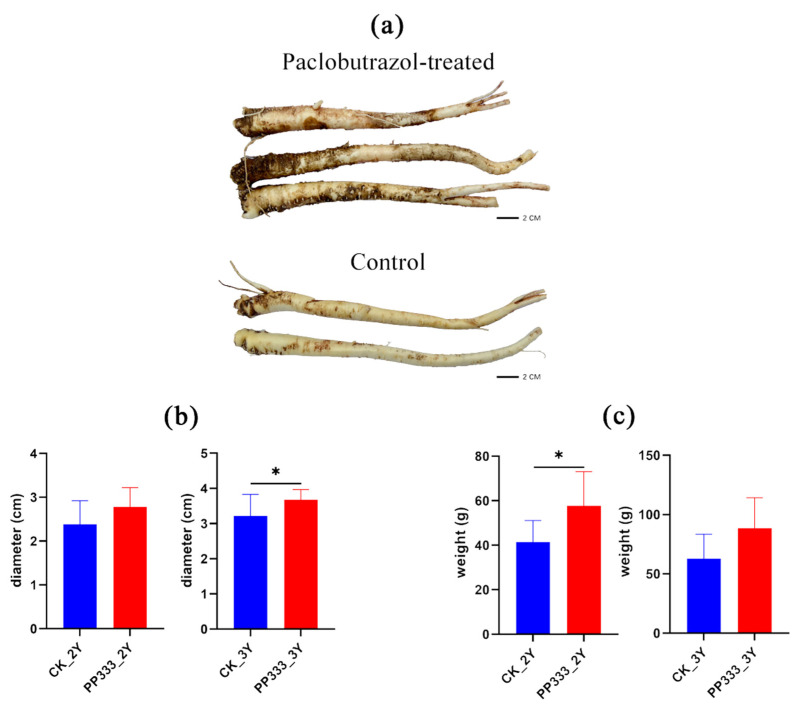
Influences of paclobutrazol (PP333) on root diameter. (**a**) Diameter (**b**) and weight (**c**) of Platycodon root. Data are presented as mean ± SD (*n* = 10). Significant differences were shown as * *p* < 0.05. CK_2Y and PP333_2Y represent control and paclobutrazol-treated group of 2-year-old Platycodon roots, respectively. CK_3Y and PP333_3Y represent control and paclobutrazol-treated group of 3-year-old Platycodon roots, respectively.

**Figure 2 molecules-27-06902-f002:**
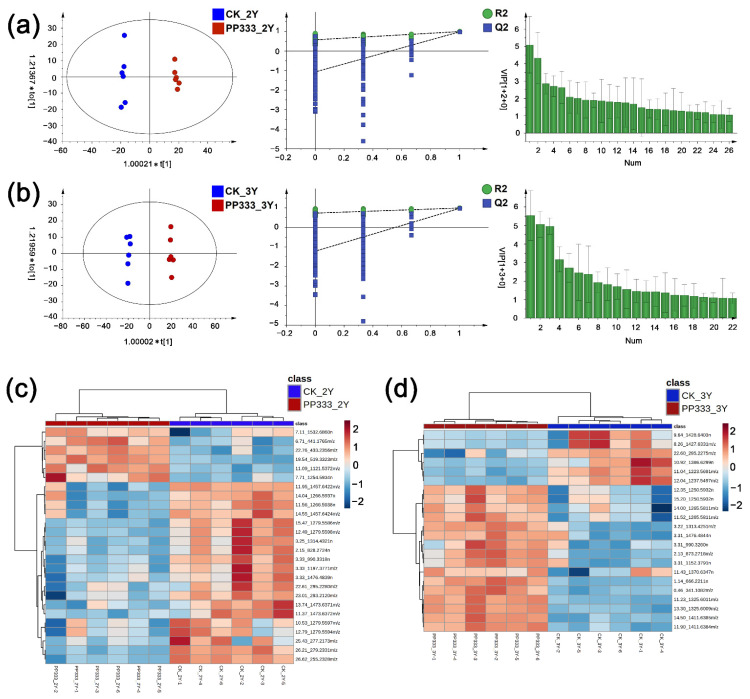
OPLS-DA score plots, permutation tests, VIP plots and heatmaps of the metabolome analysis in 2-year-old (**a**,**c**) and 3-year-old (**b**,**d**) Platycodon roots.

**Figure 4 molecules-27-06902-f004:**
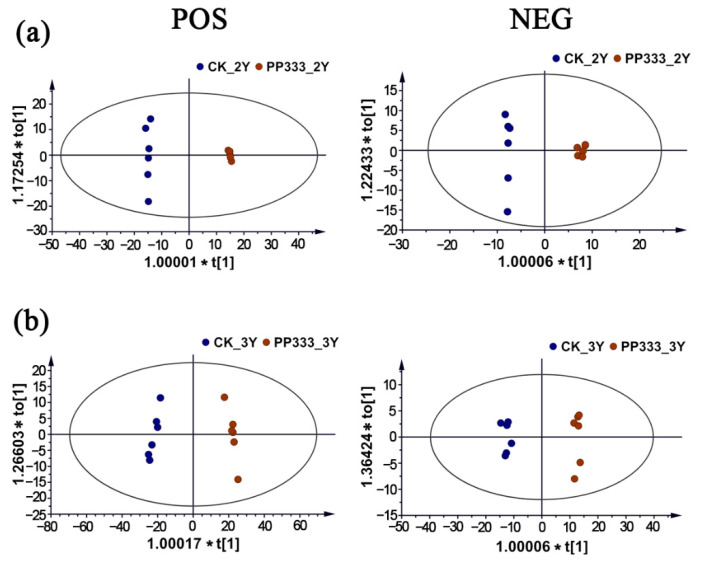
OPLS-DA score plots of the lipidomic analysis in 2-year-old (**a**) and 3-year-old (**b**) Platycodon root.

**Figure 5 molecules-27-06902-f005:**
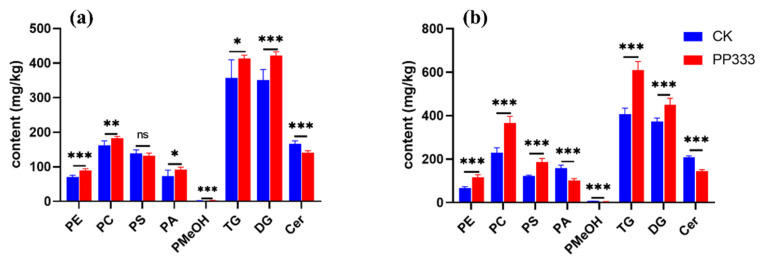
Level of lipids in 2-year-old (**a**) and 3-year-old (**b**) paclobutrazol (PP333)-treated and untreated Platycodon roots. Each value is shown as mean ± SD (*n* = 6). * indicates significant difference *p* < 0.05. ** indicates significant difference *p* < 0.01. *** indicates significant difference *p* < 0.001. ns: no significance.

**Figure 6 molecules-27-06902-f006:**
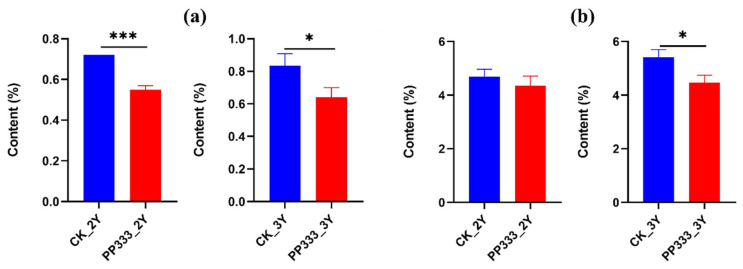
Content of saponins determined by UHPLC-ELSD (**a**) and UV (**b**) of the control and treated samples. Each value is expressed as mean ± SD (*n* = 6). * indicates significant difference *p* < 0.05, *** indicates significant difference *p* < 0.001.

**Table 2 molecules-27-06902-t002:** Effect of paclobutrazol on polysaccharides and oligosaccharides.

Content	Control		Paclobutrazol-Treated	
	2-Year-Old	3-Year-Old	2-Year-Old	3-Year-Old
polysaccharides	11.62% ± 2.03	9.40% ± 0.58	11.36% ± 1.33	11.52% ± 1.84
oligosaccharides	8.93% ± 0.85	9.41% ± 0.50	8.71% ± 0.48	8.91% ± 0.58

Notes: Results are expressed as mean ± SD (*n* = 6).

## Data Availability

Not applicable.

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
