# Peer review of "Integrated Metabolome and Lipidome Strategy to Reveal the Action Pattern of Paclobutrazol, a Plant Growth Retardant, in Varying the Chemical Constituents of Platycodon Root"

_molecules, 2022, doi:10.3390/molecules27206902_

Round 1

Reviewer 1 Report

Dear Editors, dear authors,

the manuscript

"Integrated Metabolome and Lipidome Strategy to Reveal the Action 

Pattern of Paclobutrazol, a Plant Growth Retardant, in Varying the Chemical 

Constituents of Platycodon Root"

represents an interesting and solidly presented review paper. After revising it according to my minor comments below, I recommend for publication.

COMMENTS

Line 17

in paclobutrazol treated sample --> in a with paclobutrazol treated sample

Line 19-20

explaining the exogenous matter influence --> explaining that the exogenous matter influences

Figure 2

Increase font sizes and image resolution

Table 1

How was the identification of compounds done? By which software? By database matches?

Line 125

positive and negative mode --> positive and negative ionization mode (I believe this is meant here)

Line 137

Lipid is one of --> Lipids are one of 

Line 166

How are total saponins defined?

Line 186

Similarly, polysaccharides and oligosaccharides represent a group of compounds. How were they defined here?

Line 257

Please give a reference

Reviewer 2 Report

This manuscript evaluated effect of Paclobutrazol, a Plant Growth Retardant, in chemical composition of Platycodon Root by metabolomics and lipidomics approaches. Authors comprehensively compared amounts of metabolites and lipids using UPLC-IM-QToF-MS. Additionally, using UV and ELSD, total amounts of saponin and polysaccharides were compared. Overall, the study is very interesting however some of the experimental procedure was not explained in detail. Additionally, there are some grammatical errors which requires authors to receive English editing from native speakers.

Here are some of my points.

1. In metabolomics study, author did not mention about the quality control method during the analysis. They did not mention usage of internal standard or normalization strategies to control the sample variabilities which could occur during the analysis procedure. Please mention about the method used to increase the quality of metabolomics result. Additionally, in Figure 3 authors compared intensities of total saponin based on types. Please describe how the total abundance has been calculated. If it is just sum of intensities of saponin peaks, it may not be right way to compare the level of total saponin based on each class, since each compounds have different sensitivities when analyzed by mass spectrometer.

2. Authors compare the total saponins by HPLC-ELSD and they did not describe the method but just referring to EP 10.0. Please describe which solvent condition used, and which standard is used for the quantification of total saponin, since some of readers may not have access to European Pharmacopoeia. Same for Oligosaccharide UHPLC-ELSD method. They mentioned previous method regarding calibration curve, but they did not mention about which mobile phase and analytical column has been used.

3. In Table 2, authors compared content of control and paclobutrazol treated groups. This table seems to compare control with 2-year and treated group with 3-year, which is not comparing the plant with same age. If this is the case, result may be misleading by doing wrong sampling. Please check again.

Minor points

-Page 3, Line 80: Figure 1 caption; root diameter (a) and weight (b) à root diameter (b) and weight (c)

-Page 3, Line 93: In consequently à Collectively,

-Page 3, Line 99: rising contents à Rephrase the sentences and words

-Page 3, Line 113: “type” is redundant

Round 2

Reviewer 2 Report

Authors have clearly responded to reviewer's comments.

There is one minor suggestion to authors.

Authors mentioned usage of QC samples to control the variability during analysis. I would recommend authors to add PCA plot with QC samples in supplementary files to prove the validity of analysis.
